# Social and healthcare-seeking experiences of people affected with lymphedema in Bangladesh

Kamrun Nahar Koly[1]*, Jobaida Saba[1], Zinnatun Nessa[1], Farhana Rahman Luba[1], Irin Hossain[2,3‡], M M Aktaruzzaman[3‡], Md. Golam Rabbani[4‡], Milena Simic[5], Laura Dean[6], Julian Eaton[6]

1 Health System and Population Studies Division, International Centre for Diarrhoeal Disease Research, Bangladesh (icddr,b), Dhaka, Bangladesh, 2 Department of Occupational and Environmental Health (OEH), National Institute of Preventive & Social Medicine, Mohakhali, Dhaka, Bangladesh, 3 Directorate General of Health Services, Ministry of Health & Family Welfare, Dhaka, Bangladesh, 4 Public Health Foundation, Dhaka, Bangladesh, 5 Centre for Global Health & Intersectional Equity Research, University of Essex, Colchester, Essex, United Kingdom, 6 Liverpool School of Tropical Medicine, Liverpool, United Kingdom

☯ These authors contributed equally to this work.
‡ These authors also contributed equally to this work.
* koly@icddrb.org

## Abstract

### Background

Neglected tropical diseases (NTDs) such as lymphatic filariasis (LF) are a significant concern in developing countries like Bangladesh. Understanding the health and social needs of individuals with LF is essential for improving their healthcare-seeking experiences and advancing the health system's capacity. Therefore, this qualitative study aimed to explore the social and care-seeking experiences of persons with LF in Bangladesh.

### Method

Semi-structured, face-to-face, in-depth interviews were conducted with people with LF in two highly endemic districts in northern Bangladesh. Online key informant interviews were also conducted among stakeholders associated with NTD care. Recorded interviews were analysed using thematic analysis.

### Result

28 participants (20 with LF and 8 stakeholders) were interviewed, and five major themes emerged after analysis. In terms of disease-related knowledge, the participants perceived lymphedema as a condition characterised by swelling, pain, and fever, which they believed was caused by eating stale food, being infected by others, or being punished by God. Overall, females in particular shared their experiences of negative attitudes from the community. LF adversely affected their daily life, including

**Data availability statement:** The full data of this study is not available in a public repository due to ethical restrictions. The data is very sensitive since the participants are marginalised due to their health conditions, and associated disabilities and their belonging to a already disadvantaged marginalised populations. The study involved in-depth and key informant interviews with participants. Despite the data being anonymised to protect the participants' confidentiality and the fact that consent was obtained after describing measures to protect identity, there remains a concern that people might be identifiable due to the details of the interviews. The study was conducted with the support of the Directorate General of Health Services of the Ministry of Health and Family Welfare, Bangladesh, and the Public Health Foundation, Bangladesh, which holds the data. In case of a request for any pertinent data during peer review, we will request you to contact Professor Sharmeen Yasmeen, Chair of the PHFBD-ERC, Public Health Foundation, Bangladesh (irb@publichealthfoundation.org.bd).

**Funding:** The author(s) received no specific funding for this work.

**Competing interests:** The authors have declared that no competing interests exist.

mental health and well-being. Most respondents sought support from conventional healthcare services; however, their perception of incurable disease led to low medication adherence and dissatisfaction. Lack of knowledge, inaccessibility of healthcare services, financial challenges, and physical disability were major barriers to seeking care. Participants emphasised the importance of financial assistance, community awareness, enhancing the accessibility and quality of care, and occupational rehabilitation scopes with governmental aid.

## Conclusion

Our findings highlighted the importance of ensuring an accessible and affordable healthcare infrastructure for people with LF. Additionally, the involvement of government and related stakeholders is essential to improve service users' experiences and attain high standards, combined with the need for inclusive well-being-related services. Concentrated efforts should be made to design culturally acceptable interventions to raise awareness and reduce stigma.

## Author summary

LF is one of the significant NTDs, and if left untreated, it can lead to severe complications such as lymphedema. As a leading cause of disability, it results in restricted mobility, loss of income, impaired well-being and social stigma. Even after eliminating the condition in 2023, over 70 million people remain at risk in high-endemic areas of Bangladesh. This study was conducted in two districts with high prevalence rates of LF in northern Bangladesh to explore healthcare-seeking behaviour, stigma, and challenges faced by people with LF. In-depth interviews and key informant interviews were conducted with individuals with LF and key stakeholders. Most affected individuals delayed seeking care or could not afford formal care; others relied on traditional healers. Females with LF encountered heightened stigma and exclusion, leading to emotional distress. The key challenges identified in this study included lack of awareness, limited access to healthcare, and the financial burden of treatment. Our findings highlighted the importance of providing accessible, available, and affordable healthcare infrastructure, as well as the need for inclusive well-being-related services for people with LF. A multi-sectoral approach is needed to design culturally acceptable interventions to ensure holistic care and support for LF.

## Introduction

Lymphatic filariasis (LF) is one of the most prevalent neglected tropical diseases (NTDs), affecting over 120 million people worldwide, and is responsible for the

highest health and socio-economic burden among NTDs. It is the second leading reason for physical disability and causes 5.25 million disability-adjusted life-years (DALYs) with a $5.8 billion economic burden worldwide [1,2].

Parasites causing LF are transmitted through mosquito bites, typically acquired in childhood [1–5]. If left untreated, the condition can lead to chronic manifestations, such as lymphedema [3]. These visible swellings in limbs, groin in men, and, occasionally, breasts in women worsen through painful episodes of acute dermatolymphangiodenitis, commonly known as acute attacks, caused by poor skin care. Advanced lymphedema results in limited mobility, decreased productivity at home and work, and, consequently, loss of earnings economically [6,7]. While acute attacks at any stage of lymphedema require complete rest, medication, and social support for at least 1–2 weeks at a time, putting further strain on the whole family [8].

Apart from the immediate impact of LF on the person and families, visible disfigurement can also lead to social exclusion and mental distress, impacting overall well-being [9]. A recent study reported a high burden of mental health conditions among persons affected by LF (12.6% to 71.7%), with a greater tendency towards suicidal ideation (18.5%) [10]. Different factors, such as adverse socio-economic and environmental aspects related to the wet season, were found to be associated with morbidity in LF [11]. Moreover, people with LF are often avoided by their community, and this was one of the significant social impacts of LF, as reported by Zeldenryk et al. in a critical review [12]. Additionally, studies from low- and middle-income countries suggest that experiencing such stigmatised attitudes has a bidirectional impact on increasing disease burden and worsening mental health, further acting as a significant barrier to healthcare-seeking behaviour [13].

Health-seeking behaviour is a complex concept that can be defined as the steps taken by an individual who perceives a need for help in attempting to solve a health problem [14,15]. Cultural beliefs, socio-demographic status, women's autonomy, physical and economic accessibility, and health system structure can lead to poor utilisation of healthcare services [9]. In addition to these factors, healthcare seeking for NTDs, such as filariasis, can also be compromised due to the power dynamics within communities and the disempowerment of affected individuals, which deprives them of decision-making capacity in their households [16]. Moreover, in developing countries like Bangladesh, out-of-pocket household health expenditure (63.3%) is the dominant source of health financing, followed by government subsidies; therefore, most people with LF could only receive partial care or not use formal healthcare at all [17]. Evidence supports that people with NTDs are reluctant to seek treatment, further intensifying their diseases [13]. This delayed help-seeking behaviour potentially reduces treatment efficacy, leading to sub optimal recovery rates, sustained impairment, and increased visibility of the illness, thus exacerbating negative stereotypes and discrimination in society [13]. As a result, people suffer from severe mental distress, such as suicidal ideation and depression, and negative emotions like shame, fear, and stress [13]. Therefore, delayed and sub-optimal care-seeking leads to a vicious cycle that intensifies the burden of disease, exacerbates stigmatisation, and cumulatively negative influence has a cumulatively negative influence on poor mental health.

While worldwide studies have explored the social and healthcare experiences of persons with LF, there is a dearth of evidence in Bangladesh [12]. Although Bangladesh eliminated LF as a public health problem in 2023, over 70 million people remain at risk of developing LF, especially in high-endemic areas of the northern region [3,18]. Only one study was conducted to operationalise and strengthen the quality of life tool for persons with LF, which provided some environmental and social challenges faced by those with LF [17]. However, a more comprehensive presentation of familial, social, and healthcare-seeking practices and experiences is crucial to influencing policymakers to support evidence-based interventions that incorporate a rights-based approach. Therefore, this study aims to address this critical evidence gap by exploring the pattern of healthcare seeking, experiences of stigma, perceived challenges, and recommendations from the daily lives of PWLF in two high-prevalence districts in Bangladesh, together with the accounts from healthcare providers.

## Methods

### Study setting

Participants for this study were purposively recruited from the NFEP (National Filariasis Elimination Program) database of individuals with LF, enlisted from two high-prevalence districts in northern Bangladesh: Thakurgaon (TKG) and

Panchagarh (PGR), approximately 300 km and 417 km from the capital, Dhaka, respectively. These areas were selected because they are designated as high-endemic zones due to their socio-economic status, remoteness, and proximity to the country's geographical border. Each zone has received 5–12 rounds of Mass Drug Administration (MDA) [19]. Both of these districts are under the Rangpur Division (first-level administrative unit formed by several districts), which holds the highest number of lymphedema (26,681) and hydrocele (11,661) cases [20]. Around 80% of people in these areas depend on agriculture [20]. NFEP provides free care, including diagnosis and prescriptions, through its services.

## Study design

The study employed a qualitative approach as the most effective means of gathering detailed information that would inform subsequent efforts. A semi-structured interview format, allowing the addition of open-ended responses, was used for in-depth interviews (IDIs) with people with LF and Key Informant Interviews (KIIs) with NTD stakeholders [21]. Two types of interview guidelines were developed which were informed by the literature and socio-ecological model (SEM) of healthcare-seeking behaviour [22], leading to themes related to knowledge gaps, misconceptions, incurability beliefs [23]; emotional distress, and mental health impact [24]; gender-specific challenges, family support, hiding symptoms [25]; community level stigma, isolation, avoidance of gatherings [26]; barriers to access: distance, cost, medication shortages [27]; and need for financial assistance, disability recognition and systemic neglect [28,29].

We attempted to identify various factors that influence the care-seeking pattern of individuals with LF. Our interview guides (S1 File) included questions about participants' understanding of LF [30], experiences of stigma after being diagnosed with LF [17], treatment-seeking pathways, the impact of LF on personal, professional, and social life, knowledge about mental health related to NTDs. The interview guideline for persons with LF also included questions about their understanding of available services and perceived challenges LF faces when seeking and utilising healthcare services. On the other hand, the guides for NTD stakeholders (S2 File) addressed the current situation of LF burden in Bangladesh, community perceptions and beliefs about LF, health-seeking behaviour, treatment access and barriers, mental health and psychosocial impact of LF, and programmatic support for individuals with LF. We conducted face-to-face IDIs with individuals with LF, as this approach allowed for a deeper connection between the data collectors and respondents, facilitating an extensive exploration of the subject matter while giving the researcher full control over communication to understand complex research questions through verbal and non-verbal cues [30]. On the other hand, online KIIs interviews were conducted with the NTD stakeholders since it was more convenient to the cost, time, distance, and unpredictable circumstances (e.g., weather issues) and provided additional flexibility in scheduling to maximise the chances of successfully gathering information in a way that is suitable for both the researchers and participants [31]. This study adhered to the Consolidated Reporting Criteria For Qualitative Studies (COREQ) (S3 File).

## Data collection

Potential adult participants from Thakurgaon and Panchagarh were recruited from the NFEP (National Filariasis Elimination Program) database of persons with LF. We were provided with medical records (which provided information related to the time since diagnosis or duration of the disease and treatment seeking gap, meaning diagnosed, not on treatment. We determined the eligibility of the participants based on two criteria: a. the patient still resided in the study area, and b. agreed to and was able to participate. The interviewer communicated with the persons with LF over the phone for an initial engagement. KII participants were purposively selected from the lead author's relevant professional NTD networks based on their involvement in NTD and LF-related health service provision, research, and healthcare intervention implementation programmes for at least 2 years.

All interviews were scheduled at a time, place, and mode that were convenient for the participants. For the people with LF who agreed to participate in the initial phone call, the data collectors visited their homes to conduct face-to-face IDIs. While conducting IDIs, the data collectors used recorders to audio-record the interviews for further analysis. The interviews lasted 20–40 minutes on average.

Before online KIIs, NTD stakeholders were invited to participate in the study via telephone or email. All the participants consented to participate in KIIs (considering their busy schedules) and were sent individual Zoom (online meeting platform) invitations. The KIIs were audio-recorded through Zoom software and lasted 30–40 minutes on average. The lead authors (KNK and JS) conducted all the interviews with the support of the research team, including organising and scheduling interviews, inviting participants, and recording the meetings. Both face-to-face and online interviews were conducted in accordance with the feasibility and preferences of participants and researchers.

## Analysis

The thematic analysis method, as proposed by Braun and Clarke [32], was employed for data analysis, utilising a sequential combination of deductive and inductive approaches. The research team was involved in coding data, grouping codes into themes and subthemes based on the literature and previous insights, and iteratively refining them to an agreed-upon, coherent narrative [32].

Data analysis was done by the research team members (KNK, JS, FL, ZN), who are public health graduates with prior experience with qualitative data collection and analysis. The lead author and other co-investigators transcribed and checked the audio-recorded interviews. Three experienced research team members (JS, ZN, FL) prepared a list of codes based on previous literature (deductive coding) and also performed coding independently (inductive coding). During regular meetings, codes were iteratively reviewed, reconciled, and aligned with relevant quotes. A final codebook (S4 File) was created under the supervision of the lead authors, and representative quotes were translated accordingly. All team members inspected and consented to the quotes and contextual text to guarantee reflexivity in reporting and interpretation. Initially, the intention was to include three districts (Thakurgaon, Panchagarh, and Nilphamari), but data collection was guided by the principle of thematic saturation. After conducting 20 interviews across Thakurgaon and Panchagarh, as well as 8 stakeholder interviews, thematic saturation was achieved, with no new significant themes emerging. Consequently, we concluded that data collection at that point and interviews were not conducted in Nilphamari or with additional stakeholders, as further data were deemed unlikely to yield novel insights relevant to the study objectives.

## Ethics statement

This study protocol was reviewed and approved by the Institutional Review Board (IRB) of the Public Health Foundation, Bangladesh Ethical Review Committee (PHFBD-ERC) (PHFBD-ERC-SF03/2024). The authors adhered to the National Disability Authority (NDA) and the Convention on the Rights of Persons with Disabilities (CRPD) throughout the process. Before the interview, the interviewers obtained informed consent, ensuring that participation was voluntary. Written and verbal consent were obtained from the participants with LF. Moreover, if persons with LF had any form of disability in their upper limbs or were unable to read and write, we obtained verbal consent. Additionally, we obtained thumbprints in the presence of a witness (a family member).

Additionally, NTD stakeholders provided verbal consents during their online interviews. All participants were informed about the study's objectives and purpose in simple language (Bangla) before data collection. Participants were granted complete control to halt their interviews or discussions at any point in time. They were also requested to provide consent for audio recording of the interviews. Additionally, psychosocial support sessions and referrals to the National Institute of Mental Health (NIMH), a tertiary-level public institute for mental health, were offered to individuals affected by LF.

## Study partners

This qualitative study was conducted in collaboration with the National Filariasis Elimination Program (NFEP) of the Directorate General of Health Services (DGHS) under the Ministry of Health and Family Welfare (MoHFW) of Bangladesh.

## Results

A total of 28 participants were interviewed between January and March 2024: 20 People with LF and eight stakeholders. Out of 52 on the NFEP patient list, eight had moved out of the area, 14 were unwell, and 10 could not establish a meaningful conversation.

The participants affected by LF (N = 20) (PWLF) were between 40 and 63 years old (Table 1). Among them, nine were male and 11 were female, representing the gender split in Bangladesh. All participants had lymphedema, with the condition's duration ranging from 5 to 20 years. NTD stakeholders (n = 8) were national programme consultants of a local non-profit organisation, upazila (sub-district), field assistants, health and family planning officers, representatives of an international non-governmental organisation, and researchers (Table 2).

The findings from the interviews were organised into five major themes (Fig 1): (1) knowledge of LF among affected individuals, (2) social experiences of living with LF, (3) impact of LF on overall well-being, (4) healthcare-seeking behaviour and related challenges, and (5) perceived solutions.

### Theme 1: Knowledge of LF among affected individuals

Filariasis was locally known as "god rog" ("god" = pain and "rog" = disease). Most participants with LF shared being unaware of the disease condition when they became affected for the first time. Regarding the cause of the disease, most participants were unable to clearly explain how they initially contracted LF. However, a few referred to LF as an illness spread by mosquitoes. Some of them claimed it was contagious, whereas others said that consuming expired or allergic

**Table 1. Socio-demographic characteristics of the study population (people with LF).**

| SI no: | Patient ID | Gender | Age (in years) | Duration of condition (in years) | Affected body part | Area of residence |
|---|---|---|---|---|---|---|
| 1 | PGR-IDI-01 | Male | 55 | 15 | One leg | Panchagarh |
| 2 | PGR-IDI-02 | Female | 50 | 10 | One leg | Panchagarh |
| 3 | PGR-IDI-03 | Female | 62 | 15 | Both legs | Panchagarh |
| 4 | PGR-IDI-04 | Female | 47 | 20 | Both legs | Panchagarh |
| 5 | PGR-IDI-05 | Female | 50 | 20 | One leg | Panchagarh |
| 6 | PGR-IDI-06 | Male | 59 | 10 | One leg | Panchagarh |
| 7 | PGR-IDI-07 | Male | 50 | 20 | One leg | Panchagarh |
| 8 | TKG-IDI-01 | Female | 56 | 15 | Both legs | Thakurgaon |
| 9 | TKG-IDI-02 | Female | 60 | 20 | Not specified* | Thakurgaon |
| 10 | TKG-IDI-03 | Male | 55 | 15 | Ankles of both legs | Thakurgaon |
| 11 | TKG-IDI-04 | Male | 50 | 20 | Not specified* | Thakurgaon |
| 12 | TKG-IDI-05 | Female | 42 | 11 | Both legs* | Thakurgaon |
| 13 | TKG-IDI-06 | Male | 54 | 14 | One leg | Thakurgaon |
| 14 | TKG-IDI-07 | Female | 59 | 17 | One leg | Thakurgaon |
| 15 | TKG-IDI-08 | Male | 63 | 15 | Not specified* | Thakurgaon |
| 16 | TKG-IDI-09 | Female | 50 | 20 | Not specified* | Thakurgaon |
| 17 | TKG-IDI-10 | Male | 45 | 15 | Not specified* | Thakurgaon |
| 18 | TKG-IDI-11 | Female | 40 | 5 | One leg* | Thakurgaon |
| 19 | TKG-IDI-12 | Female | 45 | 6 | Not specified* | Thakurgaon |
| 20 | TKG-IDI-13 | Male | 60 | 20 | One leg* | Thakurgaon |

*"Not specified" depicts that the participants did not specify which body part was affected by LF.

**Table 2. Socio-demographic characteristics of study population (NTD stakeholder).**

| Sl no: | Participants ID | Gender | Position | Field of Expertise | Work Experience |
|---|---|---|---|---|---|
| 1 | NTD_SP_01 | Male | Technical research consultant, non-profit health research institute | NTD and LF | 13 years |
| 2 | NTD_SP_02 | Female | Representative (communicable disease elimination programme) of an international non-governmental organisation | NTD and LF | 23 years |
| 3 | NTD_SP_03 | Male | Physician based at a governmental primary healthcare facility at Panchagarh | Health service provider of LF affected individuals | 7 years |
| 4 | NTD_SP_04 | Male | Research Consultant, Directorate General Health Services, Ministry of Health | NTD, vector-borne diseases and LF | 20 years |
| 5 | NTD_SP_05 | Male | Researcher (Parasitology), Public Health Research Institution | NTD, vector-borne diseases and LF | 9-10 years |
| 6 | NTD_SP_06 | Male | Physician and Researcher (studying vector-borne diseases), Public Health Research Institution | NTD, vector-borne diseases and LF | 8 Years |
| 7 | NTD_SP_07 | Male | Community-based health worker, International Non-Profit Organisation | LF and NTDs | 4 years |
| 8 | NTD_SP_08 | Male | Physician based at a governmental primary healthcare facility | Health service provider of LF-affected individuals | 12 years |

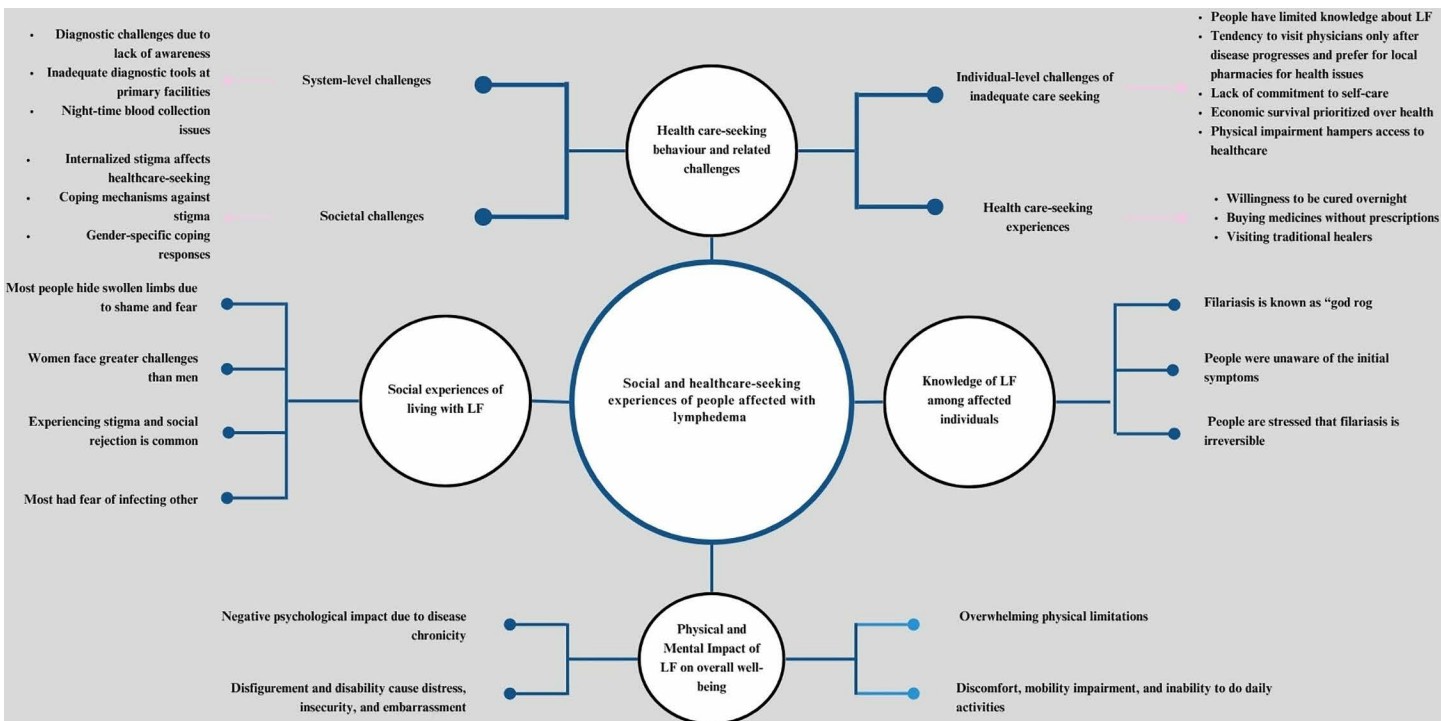

**Fig 1. Thematic distribution of major findings.**

foods was the cause of their pain and swelling. Another participant thought the pain and swelling in his leg were due to rickshaw (three-wheeled cart) pulling. Only one participant believed the Almighty had punished him for the wrong deeds of his ancestors. Additionally, most people thought the condition had no cure. As one of them stated,

*I have no idea how I contracted Filaria. Eating fish, beef, and stale food aggravates it, so I had to give up on a lot of food. Many people say it is contagious and not curable. (male, age 50)*

In summary, most participants' knowledge about the disease was related mainly to the symptoms they experienced after being affected by LF. The common symptoms were itching, rashes, fever, pain, white discharge from the affected area, and swelling in the lower limbs. A couple of participants claimed that the symptoms were exacerbated during the winter season.

*I didn't know what LF was. I previously used to see people with swollen legs. When I was affected, I first noticed pain in my arms, which proceeded toward my legs. Gradually, my legs started swelling up with additional abdominal pain and itching. (female, age 60)*

**Theme 2: Social experiences of living with LF**

Participants discussed the social experiences and challenges they and their caregivers faced. We found three drivers of stigma: appearance, community beliefs, and misconceptions. Internalised stigma led individuals with LF to hide their symptoms and avoid social interactions, impacting family relationships, especially for women. Although they received support from caregivers in daily tasks and healthcare, economic hardships exacerbated the sense of burden due to their condition. Most participants received uncomfortable stares and were avoided by their community, causing them to feel shame and embarrassment.

*Sometimes, I feel my neighbours stare at my swollen leg and talk ill about me in my absence. They interact with me when necessary, but it is minimal compared to my other family members. Thinking about it makes me sad, knowing that others judge or avoid me because of my swollen leg. (female, age 50)*

Similarly, the NTD stakeholders (Stakeholder Participant-SP) highlighted that persons with LF often hide their lymphoedema for many years, trying to conceal their symptoms, as having LF would bring abandonment and shame to their families. Concealment was explained as resulting from unpleasant bodily changes and a fear of infecting others.

*The majority of the individuals fear sharing their illness with their family members and community. For example, if a woman's foot is swollen, she covers it with a saree [traditional bangladeshi women's garment] as long as she can so that no one can notice her infection. They fear social rejection and negligence due to their condition. (NTD_SP_03)*

On the other hand, external stigma was also reported by some participants with LF and NTD stakeholders, who noted that people with LF tended to avoid local gatherings due to their fear of being humiliated. Additionally, they perceived that their condition could lead to negative attitudes among community members towards their family members.

*Most of the time, I avoid social events, weddings, and invitations. Even as a mother, I did not attend my daughter's marriage ceremony or visit her in-law's house. My appearance is not presentable, as my legs make me look ugly. It breaks my heart, but I chose not to visit them because I don't know how her in-laws would feel or react seeing my condition. (female, age 40)*

NTD stakeholders highlighted the negative experiences and stigmatisation that even extend to interactions within the family, where the health condition negatively impacts family dynamics and marriage, with women facing comparatively more obstacles than men, exacerbating their challenges further.

*People with lymphedema and hydrocele often experience different challenging situations at the family level. In many instances, some lose their relationship with their spouses as they perceive the condition might lead to infertility. Also, sometimes, people with LF, especially females, often remain unmarried. (NTD_SP_05)*

Participants with LF mentioned needing assistance regarding daily activities such as cooking, eating food, and using toilets, mainly when their pain was intense. Moreover, they also received support while accessing healthcare services and maintaining self-care related to lymphedema.

*I have a son who does everything for me in the absence of my late husband. My son ensures I take my medicines regularly and accompanies me during hospital visits. (female, age 50)*

### Theme 3: Impact of LF on overall well-being

Participants highlighted the physical limitations imposed by LF, which take a toll on emotional well-being. The findings generated two sub-themes: (1) Physical disability due to LF and (2) Adverse effect on mental health.

**Physical disability due to LF.**  Both people with LF and NTD stakeholders underscored the overwhelming physical limitations imposed by LF, with lymphedema causing discomfort, significantly impacting mobility and activities of daily living. For individuals experiencing advanced stages of the condition, movement becomes particularly challenging. The swelling and pain associated with LF hinder their ability to walk comfortably or efficiently, with many resorting to using walking aids to provide support and alleviate some discomfort.

Some people with LF reported having to cease working due to the condition. However, most individuals lived in extreme poverty, but despite this, men had to work as rickshaw pullers, farmers, household helpers, and groundskeepers despite having debility, and women had to perform all the usual household chores. However, participants perceived themselves as a burden to their families, as their economic contribution was minimal.

*I cannot earn a living because my siblings and sons do not assist me, as they have their own families to care for. I try to work as a groundsman; however, I am unable to fulfill my responsibilities within the family. I am a burden to my family (male, age 55)*

**Adverse effect on mental well-being.**  Individuals with LF become hopeless after dealing with the disease for a significant period. NTD stakeholders agreed that LF may negatively impact psychological well-being due to its chronic nature.

*Reflecting on my life before contracting the LF infection, I feel extremely overwhelmed. I cannot move independently due to the continuous pain and the never-ending discomfort, which are affecting my mental peace. (male, age 62)*

The majority of the people with LF described their mental health by saying "mon kharap" (sadness) due to their struggles to perform daily life activities and engage socially. The stigmatising attitudes of the community and physical discomfort lead to low mood and distress. Moreover, due to their low socio-economic status, they were concerned about the treatment cost and financial security of their families, having lost their jobs. Such sadness discouraged them from maintaining self-care.

*I consistently feel sad and frustrated, which began when I first learned that I had LF. When the pain aggravates, the whole legs seem swollen and reddish. When the pain becomes unbearable, it reduces my energy to such an extent that I cannot even prepare my food or go outside. I can't even take care of my legs. (female, age 47)*

Additionally, NTD stakeholders emphasised that disfigurement, disability, and limited activity caused by LF influence individuals to feel distressed, insecure, embarrassed, and have lowered self-esteem.

*People who have LF often feel ashamed about the deformity imposed by LF. In their minds, they wonder what others might think about their appearance. They always consider themselves inferior and remain emotionally disturbed. (NTD_SP_03)*

Nevertheless, people with LF reported that they tried to calm their minds by praying and engaging in religious activities to cope with their emotional vulnerabilities.

*When I first heard about my health condition, I was heartbroken. I used to ask the Almighty why he punished me with LF and started to pray all the time. It is probably the consequence of a misdeed by my ancestors, but even if I pass away with this disease, I will accept this as the Lord's will. (male, age 50)*

**Theme 4: Health care-seeking behaviour and related challenges**

People with LF seek both formal and informal care but tend to delay seeking treatment and neglect self-care practices, exacerbating the health risks. There are multiple challenges to improving the well-being of individuals with LF, including individual, societal, and systemic barriers. These findings yielded four themes: (1) healthcare-seeking behaviour, (2) individual-level challenges of inadequate care-seeking, (3) societal challenges, and (4) system-level challenges.

***Health care-seeking experiences.*** Participants who had LF stated that understanding the symptoms took a long time. They only sought treatment when they observed significant swelling of their legs and experienced pain in their limbs. Most of them sought treatment after 2–5 years of symptom presentation. Moreover, after experiencing the LF-related symptoms for prolonged periods (15–20 years), many of those affected lost interest in seeking healthcare.

The majority of individuals with LF mentioned that they opted for formal healthcare services, such as community clinics (primary-level healthcare), sadar hospitals (secondary-level healthcare centers), and medical colleges (tertiary care facilities). Furthermore, they received medicine from community clinics through a mass drug administration (MDA) programme. Some participants sought services from private doctors and local pharmacies; only two mentioned going to traditional healers ("Kobiraj"), or veterinary doctors, and one sought support from a homeopathic doctor. However, participants argued that the treatment could only manage their pain for a short time.

Participants with LF mentioned taking prescribed oral and injectable drugs that the healthcare centres distributed free of cost through community clinics under the MDA program. Additionally, most participants reported receiving various medications and topical ointments, including antiseptic cream, paracetamol, and ointments for affected limbs, as well as hygiene supplies such as soap, towels, toilet paper, cloth bandages, sitting aids, water buckets, and shoes to support foot care. Moreover, some also described attending self-care sessions at local primary care facilities where they were guided about regular exercises, self-help physiotherapy, and personal hygiene.

In addition, according to NTD stakeholders, service users tend to be optimistic about being cured overnight. Such ideologies influenced them to buy unprescribed medicines from local pharmacies and traditional healers, NTD stakeholders added.

*When people with LF come to seek treatment at healthcare centres, they expect a speedy recovery from this debilitating condition. However, when their symptoms persisted for several years, many started consuming antibiotics without consulting a doctor. Later, some opt for local treatments, such as herbal remedies or alternative therapies, which may provide temporary relief or cost savings for a short time. However, it could lead to worse consequences for the diseases in the long run (NTD_SP_01).*

***Individual-level challenges of inadequate care seeking.*** A lack of knowledge about LF was observed as a significant impediment to participants' care-seeking behaviour. The majority agreed that they did not seek medical care because they were unaware of LF and its symptoms. Symptoms were often overlooked and frequently co-occurred with other health issues, such as fever, arthritis, and fluid retention.

*My problems started when I was 8 or 10 years old. There wasn't any treatment available back in the day. Villagers were unaware of such things, so I could not receive the care I needed. (male, age 55)*

The NTD stakeholders also affirmed that most people have limited knowledge about LF. They claimed most people could not identify their symptoms and often confused LF with simple oedema. When the swelling persisted for an extended period, they consulted healthcare professionals for a diagnosis. One UHFPO stated,

*Many people consider their filaria-inflicted swellings to be a form of general fluid retention. They expect the swelling to reduce on its own. However, when they notice swelling in their legs that persists for almost one month or more, they seek medical attention to get it diagnosed. (NTD_SP_03)*

Among those who received formal care, some mentioned not completing the treatment regime, and despite taking medicines for several years for LF, most claimed that symptoms often did not reduce. Similarly, NTD stakeholders highlighted the lack of commitment to self-care among the people affected; since they face economic hardships and want to get back to their everyday lives, they often seek immediate pain or symptom-relieving medicines. Participants highlighted that, in addition to using free medicine and hygiene supplies, they had to make out-of-pocket payments for extra medication, analgesics, or skin-care supplies from local pharmacies, which was burdensome for individuals in LF, as most of them live in poverty. Some prioritised other family expenses, such as children's education and necessities, over their healthcare needs. To them, it was unnecessary to spend money a condition that was incurable.

*We're very poor people. That is why we couldn't go to the doctor for treatment. I have limited income to raise my five children, so how can I even afford treatment? (female, age 56)*

NTD stakeholders agreed that people had no alternative but to prioritise economic survival over their well-being.

*Since most patients are poor and earn their livelihood through daily work, they often face the obstacle of not being able to afford to go to the hospital or fear that it will result in a loss of income. They worry about not receiving their daily wages for the day. (NTD_SP_04)*

Physical impairment as a consequence of LF also acted as a barrier to utilising healthcare services, restricted mobility, the inability to stand and walk due to impairment caused by lymphoedema, severe pain, and discomfort in their limbs. It was also difficult for them to visit healthcare facilities or attend any social activities when the swelling and pain increased.

*Sometimes, things may rely on their mobility. If a community clinic is nearby, they may be able to walk there. But if it is far away, it may not be possible for them to walk. (NTD_SP_01)*

***Societal challenges.*** NTD stakeholders highlighted that internalised stigma impacted healthcare-seeking patterns. Coping mechanisms to respond to stigmatising attitudes, such as the tendency to hide the symptoms or self-isolate, often lead to misdiagnosis of the condition and treatment delay. Stakeholders added that gender norms led to specific behaviours, such as women covering their swollen feet with sarees and seeking treatment for pain and fever without

informing health professionals about limb swelling. Similarly, males who had hydroceles chose to remain isolated as they felt embarrassed by their condition.

*When we provided services through the MDA programme, we encountered various challenges. Males with hydrocele tried to hide their condition and refused to socialise. They tried to interact less with LF than other people. On the other hand, females often came to facilities and reported fever or pain. When they sought treatment repeatedly, we tried to explore deeper. After several questions, they agreed they had lymphoedema and then showed us their swellings. (NTD_ST_08)*

**System-level challenges.** The distance was also perceived as another barrier to accessing healthcare services by persons with LF. Although participants mentioned receiving support from local community clinics, those who needed specialised treatment had to visit tertiary care facilities far from their homes. Such transportation expenses often posed additional out-of-pocket expenditures and acted as a barrier.

*My discomfort improved with the advice and the prescribed medicines by the doctors of Rangpur Medical College (a tertiary care facility). However, it is very distant from my house, approximately a hundred kilometres away. (male, age 50)*

Diagnostic challenges were primarily discussed by NTD stakeholders. They noted that very few cases were identified during the infection stage, leading people too often ignore the symptoms and attributes less importance due to a lack of knowledge.

*Diagnosing LF is a bit challenging. This is because, after an infection, no signs and symptoms may be observed for two to three years. The diagnostic procedure poses a certain challenge in discerning whether it primarily resulted from LF. Most people seek support for fever or slight swelling. As the fever or swelling reduces after taking medicines, they are given less importance. However, the parasite keeps reproducing, and the people seek support when the condition is much more severe than before" (NTD_ST_03)*

On the other hand, some stakeholders identified the inadequate availability of antigen tests at primary facilities as another significant challenge. This led them to refer people to district-level hospitals, which may have contributed to under-reporting. Some mentioned the selective availability of antigen test kits at hotspots, with one person stating that the government only purchased and provided test kits through MDA programs.

Moreover, the diagnosis of LF presents an exceptional challenge due to the nature of the microfilariae, as claimed by NTD stakeholders. These tiny parasites primarily enter the bloodstream at night; therefore, patients requiring LF testing must have their blood collected at night to ensure the presence of microfilariae. Hence, coordinating night-time blood collection with laboratory availability becomes a significant challenge in effectively diagnosing LF.

### Theme 5: Prospective solutions

Participants highlighted the need for a multi-sectoral approach to enhance self-sufficiency, reduce stigma and care-seeking delay, and improve the accessibility and quality of care to prevent the spread of LF. These findings led to three sub-themes: (1) Enhancing self-sufficiency through financial assistance and earning**earn** opportunities, (2) Increasing awareness and care through community programmes and technology, and (3) Strengthening the healthcare system.

**Enhancing self-sufficiency through financial assistance and earning opportunities.** Participants expressed diverse needs and ideas for future support. Some participants sought governmental financial assistance, as they were

unable to engage in conventional economic sectors. This financial assistance addresses immediate healthcare needs and helps obtain other necessary resources and services. One NTD stakeholder emphasised that empowering people to earn a living would increase self-sufficiency and integration within society.

*Previously, we provided patients with rehabilitative assistance, such as purchasing cows or vans for them. Unfortunately, we do not have enough funds. If such funds were allocated for substantial business ventures, they could make even greater progress in life. If we can allow them to be self-reliant and earn a living, a slight increase in their capability would be fostered. (NTD_SP_07)*

Some affected individuals advocated for job opportunities for their children, as they were economically dependent on them. They perceived that such opportunities could improve their financial status, care-seeking behaviour, and mental well-being. Providing disability cards for those affected by LF would ensure economic security.

***Increasing awareness and care through community programmes and technology.*** Interviewees indicated the need for community-based awareness programmes and rehabilitation services. They claimed that such interventions may eventually increase knowledge about LF, reduce the care-seeking delay, and alleviate the social burden of LF. A participant claimed,

*If mass awareness programmes could be arranged, this would have educated more people about the disease, lessened the stigma, and eventually reduced the delay in seeking care. (female, age 62)*

According to NTD stakeholders, in the context of Bangladesh, technology has become increasingly paramount as a potent means to disseminate awareness.

*Utilising mass media is crucial for raising awareness about LF. For example, people can easily access publications, various news, and TV channels. Now, people have smartphones in their hands. Especially in epidemic areas, targeting these points randomly where the number of patients is high, if these messages can be delivered. (NTD_SP_01)*

***Strengthening the healthcare system.*** Participants suggested that the government needed to increase the accessibility to healthcare services for persons with Lymphoedema; this included access to mental health services at their nearest healthcare centres to support their mental well-being. Many also expressed a desire for facilities that could provide a cure for LF.

*We're interested in going there if the government initiates any steps to enhance the treatment, such as incorporating counselling at the root level. By doing this, people will become more resilient and improve their self-care, enabling them to manage with chronic conditions effectively. (male, age 55)*

To sustain and enhance this quality of care, a stakeholder highlighted the importance of establishing dedicated spaces within healthcare facilities, ensuring direct access to essential services for LF patients. Moreover, healthcare workers should receive refresher training to maintain service quality and promote proactive engagement in community healthcare.

Some participants indicated they needed free medicines for the management of their acute infectious attacks. Although the Government of Bangladesh provided medications free of cost through the MDA programme, participants were concerned about the inconsistent supply of free medicines and supplies, as well as the fact that these were insufficient to meet their long-term needs. Additionally, when the free supplies ran out or required specialised care, participants had to incur out-of-pocket expenses to seek support from tertiary and private healthcare facilities for consultations, additional medications, painkillers, and hygiene supplies. One of them mentioned,

*I believe that if the medicines were provided free of charge, this would help us immensely in adhering to our medication regimen. (female, age 56)*

NTD stakeholders stated the need for proactive measures to identify and address LF cases beyond the hotspots. They also emphasised the importance of implementing passive surveillance and tracking systems in additional regions to detect any re-emergence of LF.

*Our concern is that, although we have identified LF cases in 19 districts, there are likely to be more districts beyond those. With people constantly moving between districts, LF transmission is a risk in areas not yet designated as endemic. Therefore, implementing passive surveillance and tracking systems in these additional districts is crucial. By closely monitoring monthly health reports, we can promptly detect any surge in LF cases outside the known areas. This approach enables us to intervene swiftly and implement effective management strategies. (NTD_SP_04)*

## Discussion

Lymphatic Filariasis has been identified as the second leading cause of disability worldwide. Although research on care pathways is available from some low- and middle-income countries (LMICs), there is a limited exploration of the lived experiences of people with lymphatic filariasis (LF) in Bangladesh, despite having several endemic zones [20,23,33–37].

This study interviewed individuals with lived experiences of LF and NTD stakeholders to understand their views on healthcare-seeking behaviour while discussing their concerns regarding various challenges to utilising healthcare services. Our study observed long-term sadness, worry about the future, and low life satisfaction among the people affected by LF. Similarly, evidence from Nigeria, Togo, and India reported that the prevalence of depression ranges from 20% to 90%, and due to its chronic manifestation with acute episodes of fever, people experience a double burden of compromised physical health and disability [38–41].

Our findings demonstrated that people affected by LF had limited knowledge about the symptoms, causation, and mode of transmission [9,42]. These findings are similar to those of a study in Zambia, where participants perceived that LF was caused by eating expired food, witchcraft, or using objects previously used by individuals affected by LF [16]. Other studies have also shown a limited understanding of LF transmission, including the role of mosquitoes as vectors. A review study found that low awareness of NTDs and their severity can lead to delays in healthcare-seeking among affected individuals, which aligns with our study [43]. We found that participants affected by LF from northern Bangladesh used local terminologies when referring to the names and symptoms of the disease. In Bangladesh, LF is locally known as "godrog", whereas in Zambia, it is referred to as "*musakasa*" (meaning swollen limbs) *and "erisipela* or *dicipela"* (bacterial skin infection) in the Dominican Republic [16,44]. Local terminologies should be considered when developing information, education, and communication (IEC) materials that positively influence the care pathway, as recommended by participants in this study [45]. Enhanced community awareness may reduce delays in care-seeking and also support the reduction of high levels of stigma within the community, although knowledge change itself is insufficient. Combining contact intervention and mass awareness can be a powerful approach to reducing stigma. Contact-based interventions, particularly indirect social contact (ISC) approaches involving active engagement, such as group discussions or problem-solving exercises based on real-life stories, are effective in reducing mental health-related stigma, especially in low- and middle-income countries [46].

Our data on participants' social experiences provided valuable insights regarding the drivers of stigma experienced by individuals with LF. While the physical manifestations of LF, particularly lymphedema, undoubtedly contribute to stigma

due to altered appearance and looks, participants' narratives suggested that a lack of accurate knowledge about LF within communities, coupled with fear regarding its potential transmission, contributed to social withdrawal and isolation among persons with LF. Furthermore, the internalisation of local beliefs, such as the perception of LF as a form of divine punishment, seemed to influence participants' self-worth and acceptance of their condition, potentially exacerbating internalised stigma. Thus, our data suggest that while physical appearance is a factor, the social experiences of individuals with LF are shaped by a complex interplay of limited knowledge, fear of contagion, and cultural and religious beliefs, all of which contribute to the overall burden of stigma. Consistent with our findings, several studies also identify fear of contagion and the inability to fulfil societal roles as significant contributors to stigma among individuals with LF [47–51].

Experiencing humiliation in the community is a common scenario among people with skin NTDs like LF, leprosy, and onchocerciasis [43,52]. This aligns with our study findings, where individuals affected by LF reported feeling shame and disgrace, as well as experiencing uncomfortable stares and avoidance from their community. Therefore, such social exclusion results in higher levels of disability, incomplete education, and unemployment, catalysing a downward spiral into poverty [38]. Community-level stigma can also directly impact mental health, cause delays in diagnosis and treatment, and consequently, the advancement of the disease [53]. A study in Togo found that over 70% of respondents exhibited a higher rate of depression according to the DUKE-AD scale (score >30), with this rate escalating as lymphoedema progressed to more advanced stages [40]. Women affected by LF living in the Dominican Republic, Ghana, and Thailand experienced more significant stigma than men, which is also observed in our study [25,54,55]. A pair of studies from Ethiopia exposed that women with LF faced divorce, were denied marriage, and missed out on motherhood [55,56]. As identified in other settings, including Ghana and Haiti [25,49,50,57], people with LF received support for self-care and accessing healthcare, which helped them cope [49,50].

Our study findings also revealed that people affected by LF received assistance with household chores, self-care management, and utilisation healthcare services from caregivers. As has been identified in other settings, including Ghana and Haiti [25,49,50,57], this support often helped affected individuals cope with the conditions and stigma from the community [49,50]. Family support should be an opportunity to ensure social support for people affected by LF and their caregivers. There is a need for future research to develop culturally contextualised family-based interventions that can combat stigma and promote mental well-being, both for people affected and carers. Thus, support for caregivers is considered essential when developing community- or family-based mental health support services.

In our study, participants sought treatment from different levels of formal and informal healthcare facilities to find cures for their chronic pain and complications. However, other studies have found that traditional healers remain the primary healthcare choice for many people with LF in Bangladesh [58]. In Sri Lanka, acupuncture and traditional remedies are commonly used in conjunction with Ayurvedic treatments for lymphoedema [59,60]. Consequently, a significant number of patients delay visiting medical facilities until their symptoms have deteriorated and pain has intensified, which adversely impacts the effectiveness of treatment [59,60]. Moreover, various studies have identified a significant decline in motivation to seek care upon realising the chronic nature of the condition and the improbability of complete recovery [44,61–63], and people do not take treatment consistently [45].

Additionally, lack of satisfaction with prescribed drugs caused low therapeutic adherence among the participants, which was similar to that observed in Ghana, where people refuse to ingest LF medicines due to the fear of adverse drug reactions and misconceptions about the condition [57]. Moreover, some studies have also discussed how experiencing stigmatised attitudes from healthcare providers impacts the healthcare-seeking attitudes of persons with LF. We were surprised that our participants did not recall facing any stigma from healthcare providers [17,64]. Such interactions are often common in particular settings where healthcare providers have limited LF-specific training, resulting in neglect, verbal judgment, or a lack of empathy. This situation reflected the recent efforts in the NFEP programme of the Bangladesh Government to improve health service delivery. However, this may also indicate that patients have low hopes of care and thus did not translate neglecting behaviour as stigma. Alternatively, participants may also have chosen not to discuss the

negative experiences due to fear, loyalty, or social desirability bias. Therefore, future studies should aim to address this gap to better understand the relationship between healthcare providers and individuals with LF in shaping care pathways.

Regarding the barriers to seeking healthcare among persons affected with LF, our study findings align with the Social Ecological Model (SEM) [22], a theoretical framework that provides a comprehensive understanding of how different factors interplay and influence people's healthcare-seeking behaviour and health outcomes. Low health literacy about LF, overwhelming emotional responses, and distress related to LF were common at the individual level, and such perceptions resulted in delayed care seeking. Interpersonal connections, defined by caregivers' involvement and family support, played a vital role in ensuring access to care. However, the gender disparity and social stigma affected females particularly and resulted in reduced help-seeking tendencies. On the other hand, the dominant role of community-level stigmas and misconceptions of LF pushed people with LF towards experiences of judgement and social withdrawal, further reinforcing delayed care seeking. At a system level, structural barriers marked by long distances, low transportation facilities, and unavailability of mental health services reduced people's dependency on the formal health system and encouraged reliance on informal care. Finally, at the policy level, participants expressed their concerns about the need for stronger governmental efforts, such as providing financial assistance or accessible rehabilitative services. Therefore, applying SEM underscores the need for integrated efforts to improve the overall care pathway.

Participants highlighted a range of needs for better support, including individual, familial, community, and system-level assistance. Firstly, financial aid is provided to those unable to work in traditional sectors, along with job opportunities for their children, emphasising the importance of self-sufficiency and empowerment. Similar findings have been reported in a Sri Lankan study, which demonstrated the effectiveness of providing direct financial support to affected families through the Sri Lankan Samurdhi poverty alleviation scheme, which offers allowances to designated low-income households [53]. Secondly, there is a call for community awareness programmes that utilise technology effectively, as well as for improved healthcare services, including mental health support and specialised care corners within hospitals. A study conducted in Nigeria also emphasised that a thorough understanding of transmission factors at the grassroots level would significantly enhance the effectiveness of LF control strategies [65]. In terms of well-being, utilising psychological peer support has been proven effective in India in mitigating stigma and improving mental well-being, and can be implemented in practice [66]. Moreover, study findings suggest that the parallel implementation of enhanced and post-elimination surveillance in hotspots and new areas is also crucial for effectively addressing the spread of filariasis. Additionally, a study from Malawi suggests that post-elimination efforts should be integrated with other national programmes and incorporated into routine health systems [28]. In light of these findings, future research should focus on evaluating the effectiveness of integrating these strategies and exploring innovative approaches to enhance community engagement and self-sufficiency in affected populations.

### Programmatic implications of the findings

The present study suggests that concentrated efforts are necessary at both the community and policy levels to design interventions that alleviate morbidity and improve the well-being of individuals affected by LF (Table 3).

### Limitations and strengths

While this study provided insights into the experience of people living with LF, the lack of an appropriate control group, either including healthy controls or people with other chronic debilitating diseases, may have reduced generalisability. This qualitative study was designed to explore the lived experiences of individuals affected by LF and related healthcare stakeholders in low-income settings. Therefore, we focused on gathering in-depth insights from the participants, which would have been difficult to achieve through quantitative analysis or comparative study [9]. We found that people affected with LF were highly stigmatised, and often subjected to discrimination based on their visible lesions or disabilities. Some aspects of their experience may be relevant but are unlikely to be easily applicable to people with other chronic debilitating

**Table 3. Programmatic implications of the findings.**

| Area | Constraints/ Barriers | Recommendations/ Opportunities for Interventions |
|---|---|---|
| 1. Challenges of seeking healthcare by the persons with LF | 1.1. Financial barriers | a. Providing direct financial assistance or building a social safety net programme to ensure financial support.<br>b. Increasing job or rehabilitation opportunities to safeguard economic independence and create pathways to contribute to the national economy.<br>c. Arranging free-of-cost medical support to reduce out-of-pocket expenditures |
| | 1.2 Lack of awareness about the condition and community-level stigma | a. Organising mass awareness programme to educate people about LF and its chronic health complications, eventually influencing treatment-seeking behaviour.<br>b. Training formal and informal healthcare providers to encourage early case detection |
| | 1.3. LF inflicted disability | a. Increasing easy access to healthcare services through digital media<br>b. Involving community level health care workers to reduce distance barriers |
| 2. Mental health difficulties | 2.1. Presence of mental distress | a. Ensuring mental health facilities at primary-level care facilities.<br>b. Creating and promoting self-help groups to increase the social activity of persons with LF.<br>c. Capacity building of caregivers and families to enhance a positive family environment.<br>d. Training of people with LF on personal hygiene and self-care activities to ensure physical and mental well-being |

diseases. Future large-scale studies focusing on quantitative data should be conducted to validate these findings and explore the social and health-care-seeking experiences of PWLF. Moreover, we intended to gather the lived experiences of LF-affected Bangladeshi individuals in this study, which was not presented in any other studies that may have introduced bias. Therefore, future studies should integrate objective measures or adopt a mixed-methods approach to gain a better understanding.

Additionally, the interviewed participants had been infected and affected by LF for many years, which increased the likelihood of recall bias. Additionally, participants' responses from persons with LF may have been influenced by the stigma associated with the condition and societal expectations of acceptability, potentially leading to social desirability bias. This bias was controlled by ensuring anonymity and confidentiality. Regarding the dropout rate of 32 out of 52 participants, nonresponse bias may have been introduced due to systematic differences based on demographic characteristics, health status, or other relevant factors, if possible. Hence, to avoid inconsistency, the data were analysed by multiple researchers to compare interpretations.

Although we have interviewed different stakeholders who were national programme consultants of a local non-profit organisation, research consultants of DGHS, representatives of an international non-government organisation, and researchers, we were unable to include perspectives of informal care providers (traditional healers, Unani, or herbal medicine practitioners), which would provide further insights into care-seeking dynamics.

To the best of our knowledge, this study is the first to document the healthcare-seeking patterns and lived experiences of individuals affected by LF. These findings could contribute to the existing literature and support relevant stakeholders in advocating for, protecting, and promoting their rights to the government.

## Conclusion

Bangladesh has eliminated LF as a public health problem [3]. To fulfil their SDG commitments to leaving no one behind through the attainment of Universal Health Coverage (UHC), services need to be accessible for those already affected. Considering that LF and other NTDs are most prevalent in poor populations, the success or failure of the UHC can be measured against the extent of its effectiveness in reaching persons with NTDs. Even after the elimination of LF, it is crucial to focus on mental health and overall well-being due to the long-term disabilities and chronic conditions associated with the disease, and such ongoing post-validation measures reduce the risk of undetected transmission, leading to a resurgence of infections to previous levels [67]. Integrating accessible support services is valuable for the health and well-being of those affected, but it can also be an essential measure in maintaining elimination successes [68].

## Supporting information

**S1 File. IDI guideline.**
(DOCX)

**S2 File. KII Topic Guide.**
(DOCX)

**S3 File. COREQ (Consolidated criteria for Reporting Qualitative research) Checklist.**
(DOCX)

**S4 File Codebook.**
(DOCX)

## Acknowledgements

The authors are grateful to all the participants who contributed to this study. The authors are also thankful to the National Filariasis Elimination Program (NFEP) of the Directorate General of Health Services (DGHS) under the Ministry of Health and Family Welfare (MoHFW) of Bangladesh for their support in project-related work.

## Author contributions

**Conceptualization:** Kamrun Nahar Koly, Irin Hossain.

**Formal analysis:** Kamrun Nahar Koly, Jobaida Saba, Zinnatun Nessa, Farhana Rahman Luba.

**Investigation:** Kamrun Nahar Koly, Jobaida Saba.

**Methodology:** Kamrun Nahar Koly, Jobaida Saba.

**Project administration:** Kamrun Nahar Koly, Irin Hossain, M M Aktaruzzaman.

**Supervision:** Kamrun Nahar Koly, Jobaida Saba.

**Writing – original draft:** Kamrun Nahar Koly, Jobaida Saba, Zinnatun Nessa, Farhana Rahman Luba.

**Writing – review & editing:** Kamrun Nahar Koly, Jobaida Saba, Zinnatun Nessa, Farhana Rahman Luba, Irin Hossain, M M Aktaruzzaman, Md. Golam Rabbani, Laura Dean, Milena Simic, Julian Eaton.

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
