## [Decision Letter · Decision Letter 0]

17 Apr 2025

Response to Reviewers
Revised Manuscript with Track Changes
Manuscript

Shaden Kamhawi

co-Editor-in-Chief

Paul Brindley

co-Editor-in-Chief

**Journal Requirements:**

At this stage, the following Authors/Authors require contributions: Kamrun Nahar Koly, Jobaida Saba, Zinnatun Nessa, Farhana Rahman Luba, Irin Hossain, M.M. Aktaruzzaman, Md. Golam Rabbani, Milena Simic, Laura Dean, and Julian Eaton. Please ensure that the full contributions of each author are acknowledged in the "Add/Edit/Remove Authors" section of our submission form.

2) Tables should not be uploaded as individual files. Please remove the table files from the online submission form. Tables should be included in your manuscript file as editable, cell-based objects. For more information about how to format tables, see our guidelines:

https://journals.plos.org/plosntds/s/tables 

3) PLOS journals require authors to make all data necessary to replicate their study’s findings publicly available without restriction at the time of publication. When specific legal or ethical restrictions prohibit public sharing of a data set, authors must indicate how others may obtain access to the data. 

PLOS strongly recommends sharing data in a repository whenever possible. Data repositories improve discoverability and accessibility, ensure long-term preservation, and lead to increased attention for the research.

Please state where the data is stored and how readers can access it. Please also provide contact information for a data access committee, ethics committee, or other institutional body to which data requests may be sent.

Note: The point of contact cannot be an individual. It needs to be an institutional or ethics board email contact.

Please update your Data Availability statement in the online submission form after redrafting it more formally and concisely as this will be published in the paper.

**Reviewers' comments:**

**Key Review Criteria Required for Acceptance?**

**Methods**

-Are the objectives of the study clearly articulated with a clear testable hypothesis stated?

-Is the study design appropriate to address the stated objectives?

-Is the population clearly described and appropriate for the hypothesis being tested?

-Is the sample size sufficient to ensure adequate power to address the hypothesis being tested?

-Were correct statistical analysis used to support conclusions?

-Are there concerns about ethical or regulatory requirements being met?

Reviewer #1: 1. Lack of a healthy control group or comparison with other chronic diseases: Without a control group (e.g., individuals with diabetes or chronic venous insufficiency), it is impossible to determine whether the impact of lymphedema differs from other chronic conditions. Additionally, the sample size is too small (only 20 patients and 8 healthcare policymakers), limiting the ability to capture variations across gender, economic status, and disease duration.

2. Overreliance on patient self-reports without objective validation: The study does not incorporate objective indicators (e.g., medical record reviews, treatment costs) to verify the accuracy of patient-reported experiences. It is essential to integrate quantitative data (e.g., medical records, prescription data, healthcare expenses) and include a broader range of stakeholders, such as community health workers, herbal medicine practitioners, and policymakers.

Reviewer #2: Introduction:

- line 81 and 82- when describing Cluex and Anopheles mosquitos, is this urban/semi-urban and rural ares of Bangledesh or globally or where? For Ades its clear its specific to the Pacific islands.

Methods:

- lines 148-150- restructure sentence. A topic guide with open ended questions is the tool for a semi-structured interview so saying 'as well as a topic guide...' does not make sense.

- Line 150-156- Make it more clear the topics covered for each of the two groups of pts.

- line 161- 'felt well enough to participate'- what does this mean? Is this well enough physically, emotionally? Does it mean 'agree to participate' which is more common wording used around inclusions/exclusion criteria.

- what was the eligibility criteria for KIIs?

- How did you consent people who were illiterate?

- provide the KII guide as a supplementary file.

- overall review and revise methods to make clear how the two different methods (IDIs and KII) were done.

Reviewer #3: The study design chosen is sound and rightly helps address the research objectives which is inline with best practices for qualitative research.

The objectives were clearly stated. The authors provided a clear explaination and a sound rationale for the study.

The desciption of the population is generally clear, however the 8 NTD stakeholders background could be clearer and specific if they are consultant for LF or NTDs in general, this will provide a better context of their experiences to readers(line 224-226)

In Table 1, can authours clarify what 'Not clear' means under the 'Affected body part'. (Is it that participants could not describe which body part is/was affected?)

**Results**

-Does the analysis presented match the analysis plan?

-Are the results clearly and completely presented?

-Are the figures (Tables, Images) of sufficient quality for clarity?

Reviewer #1: 1. Lack of a structured thematic analysis framework: While the study employs thematic analysis, it fails to define each theme explicitly, leading to largely descriptive summaries without a detailed coding framework (codebook). The study does not sufficiently explore how the disease influences patients' decision-making processes. The analysis should align interview data with theoretical frameworks such as the Health Belief Model (HBM) or the Social Ecological Model (SEM) to enhance interpretability.

2. Limited quantitative analysis of social and healthcare barriers: The study does not quantify associations between key variables (e.g., social discrimination vs. healthcare delays). It also does not use crosstab analysis to compare gender, income level, and geographic disparities in healthcare accessibility. To strengthen the findings, quantitative analyses (e.g., chi-square tests, logistic regression) should be incorporated to explore the influence of social factors on healthcare-seeking behavior.

Reviewer #2: - line 219-220- reference the previous research that is mentioned. But also isn't saturation something you determine? Did you reach saturation?

-Table 1- age and duration of condition need units

- when using quotes, usually pt IDs are provided with the profile of pts . There is also a table of pts, so one can then easily check that table to see other characteristics of the pt that said that quote if the ID is provided.

- line 234- "knowledge of LF among affected individuals and caregivers". Caregivers are not mentioned as participants. if caregivers were interviewed, this needs to be covered in the methods.

- Theme 2- did your data speak to the drivers of stigma? Often with skin conditions, its physical appearance but what role(s) do knowledge, local beliefs, fear for example also play?

- line 400- Can you provide more of an explanation around the costs incurred for LF treatment in this context. This comes out again in line 500 on solutions, 'free medicines'. But the readers need more of an understanding of the costs to patients and what is provided by the health system.

- line 412-413- hiding symptoms is not a stigmatising attitude but a coping mechanism

- was there any stigma from health workers that may be a barrier to seeking care?

- does the data speak to other coping mechanism particularly around mental health- what existing support systems do people use if at all?

Reviewer #3: Generally the authors have a sound analysis plan given that they adopted the COREQ.

In the methods the Authors described two main particiapnts for the study. However, the first major theme mention '(1) knowledge of LF among affected individuals and CAREGIVERS. The authors can provide a bit of clarity regarding source of data for caregivers experience. (line 230-231) This theme currently suggest caregiver were probably interveiwed.

Authors could help international readers to understand 'Rickshaw' pulling' by adding ('two- or three-wheeled cart').

Can authors define what SP means ((NTD_SP_03) in line

271.

On line 361-361 authors mentioned hygiene supplies such as 'antibiotic cream, paracetamol, and ointment' I'm not if these three items have been categorised appropriately. Authors should check and describe these items approapriately

line 400: 'NTD stakeholders agreed about people being forced to prioritise economic survival over their well401

being'. Can the authors provide details about how they 'Force' to prioritise....or who force them.....Maybe be rephrase..........

line 406.. do authors want to say the '.......inability to stand and walk due to..........'. Check and correct

In relation to stigma did the authors identified the two dimensions (internalised and external stigma?) as self-isolation can also be as a result of internalised stigma which makes affected persons to hide and disassociate themselves from others. If such narrative was found it will be helpful for authors to highlight this nuases in the stigma experiences to inform all encompassing stigma interventions.

Line 501-502 Is LF medications not already free in Bangladesh?(This is a rhetoric question)

**Conclusions**

-Are the conclusions supported by the data presented?

-Are the limitations of analysis clearly described?

-Do the authors discuss how these data can be helpful to advance our understanding of the topic under study?

-Is public health relevance addressed?

Reviewer #1: (No Response)

Reviewer #2: - line 564 says traditional healers remain the primary health care choice while line 357 says only 2 went to traditional healers. please clarify.

- line 575 is a separate section from previous paragraph. Make it a clear break and indicate they are study limitations.

- a large part of the results were on solutions and this should be addressed in the discussion.

Reviewer #3: The author's conclusion supports the data analysed in this manuscripts. The authors also described potential biases in the processes as limitations how also provided information on the effort made to address such limitations in order not to affect the interpretation of the findings.

The authors have also provided a sound implication of the study which contribute to the knowledge gaps in the experiences of people affected with LF and NTD in general.

Generally the findings from this research is relevant for planning for public health interventions for addressing stigma and poor health seeking behaviour.

**Editorial and Data Presentation Modifications?**

Reviewer #1: (No Response)

Reviewer #2: In the abstract, it says "about 28 participants"- why "about" rather than being more definitive on the number? Also there are two types of participants (people with LF and key informants) so when talking about participants it should be clear which of the two groups.

Reviewer #3: I am satisfied with the data presented in this manuscript

**Summary and General Comments**

Reviewer #1: This study explores the social and healthcare-seeking experiences of individuals with lymphedema in Bangladesh, using qualitative research methods (semi-structured interviews and key informant interviews). The study analyzes the challenges faced by patients in terms of disease awareness, social exclusion, economic burden, healthcare accessibility, and mental health.

However, the study has several limitations.

Reviewer #2: This is a very interesting study and important research to inform programme delivery for LF as well as considerations to tailor education and awareness of LF necessary to improve knowledge, reduce stigma and reduce effects on mental health. The paper demonstrates differences by gender which is important when considering gender specific messaging and counselling.

This manuscript can benefit from some copy editing, particularly grammatical errors and standards of scientific writing. The results, discussion and conclusion are well written while the intro and methods feel less coherent and would benefit from copy editing.

Reviewer #3: The message articulated in this manuscript is sound and clear which makes it easy to follow. The use of clear section headings make the reading and reviewing easy to navigate. While these broad thematic findings are not new in relation to what we already know with the experiences of may skin NTDs, the authors have added a contextual narratives to the literature to expand the knowledge in the field. I highly recommend it acceptance for publication.

PLOS authors have the option to publish the peer review history of their article (what does this mean? ). If published, this will include your full peer review and any attached files.

**Do you want your identity to be public for this peer review?** For information about this choice, including consent withdrawal, please see our Privacy Policy .

Reviewer #1: No

Reviewer #2: No

Reviewer #3: No

**Figure resubmission:****Reproducibility:** To enhance the reproducibility of your results, we recommend that authors of applicable studies deposit laboratory protocols in protocols.io, where a protocol can be assigned its own identifier (DOI) such that it can be cited independently in the future. Additionally, PLOS ONE offers an option to publish peer-reviewed clinical study protocols. Read more information on sharing protocols at https://plos.org/protocols?utm_medium=editorial-email&utm_source=authorletters&utm_campaign=protocols

---

## [Editor Report · Decision Letter 1]

22 Jul 2025

Dear Dr Koly,

We are pleased to inform you that your manuscript 'Social and healthcare-seeking experiences of people affected with lymphedema in Bangladesh' has been provisionally accepted for publication in PLOS Neglected Tropical Diseases.

Best regards,

Adly M.M. Abd-Alla, Prof asso.

Section Editor

Adly Abd-Alla

Section Editor

Shaden Kamhawi

co-Editor-in-Chief

Paul Brindley

co-Editor-in-Chief

---

## [Editor Report · Acceptance letter]

Dear Dr Koly,

We are delighted to inform you that your manuscript, "Social and healthcare-seeking experiences of people affected with lymphedema in Bangladesh," has been formally accepted for publication in PLOS Neglected Tropical Diseases.

Best regards,

Shaden Kamhawi

co-Editor-in-Chief

Paul Brindley

co-Editor-in-Chief
